# Genetic Variations within the Bovine *CRY2* Gene Are Significantly Associated with Carcass Traits

**DOI:** 10.3390/ani12131616

**Published:** 2022-06-23

**Authors:** Xuelan Li, Enhui Jiang, Kejing Zhang, Sihuan Zhang, Fugui Jiang, Enliang Song, Hong Chen, Peng Guo, Xianyong Lan

**Affiliations:** 1Key Laboratory of Animal Genetics, Breeding and Reproduction of Shaanxi Province, College of Animal Science and Technology, Northwest A&F University, Xianyang 712100, China; lixuelan1020@126.com (X.L.); jiangenhui@yeah.net (E.J.); zkj679@126.com (K.Z.); sihuanzhang1990@163.com (S.Z.); Chenhong1212@263.net (H.C.); 2Institute of Animal Science and Veterinary, Shandong Academy of Agriculture Science, Jinan 250100, China; fgjiang2017@163.com (F.J.); enliangs@126.com (E.S.); 3College of Computer and Information Engineering, Tianjin Agricultural University, Tianjin 300380, China

**Keywords:** cattle, *CRY2*, indel, carcass traits, association

## Abstract

**Simple Summary:**

Carcass traits are crucial for beef cattle breeding. *CRY2*, as a clock gene, affects the body’s metabolism and regulates the physiological function of skeletal muscles, including muscle growth and development. In order to investigate the relationship between the *CRY2* gene and cattle carcass traits, a total of 705 individuals from the Shandong Black Cattle Genetic Resource (SDBCGR) population were randomly collected. Eight potential insertion/deletion sites within the *CRY2* gene were analyzed, and two of them were polymorphic. Population genetic parameters including genotype frequencies and allele frequencies of the two loci in the tested population were calculated. Importantly, the two loci were significantly related to carcass traits such as gross weight, ribeye, and high rib, suggesting that *CRY2* shows the potential for bovine growth and development. The two loci can be used as effective molecular markers affecting carcass traits of beef cattle and provide certain scientific and theoretical bases for practical breeding.

**Abstract:**

As an important part of the circadian rhythm, the circadian regulation factor 2 of cryptochrome (*CRY2*), regulates many physiological functions. Previous studies have reported that *CRY2* is involved in growth and development. However, the relationship between *CRY2* gene polymorphism and cattle carcass traits remains unclear. The aim of this study was to detect the possible variations of the *CRY2* gene and elucidate the association between the *CRY2* gene and carcass traits in the Shandong Black Cattle Genetic Resource (SDBCGR) population (*n* = 705). We identified a 24-bp deletion variation (*CRY2*-P6) and a 6-bp insertion variation (*CRY2*-P7) in the bovine *CRY2* gene. The frequency of the homozygous II genotype is higher than the heterozygous ID genotype in both two loci. In addition, *CRY2*-P6 was consistent with HWE (*p* > 0.05). Importantly, the *CRY2*-P6 variant was significantly associated with 12 carcass traits, including gross weight, ribeye, high rib, thick flank, etc. and the II was the dominant genotype. The *CRY2*-P7 site was also significantly correlated with five traits (gross weight, beef-tongue, etc.). Collectively, these outcomes indicated that the two Indel loci in the *CRY2* gene could be used for marker-assisted selection of cattle carcass traits.

## 1. Introduction

Beef quality is influenced by a combination of various factors, including breed, age, sex and gene genotype, as well as environmental factors such as feeding management and slaughter processes [1]. The Shandong Black Cattle Genetic Resource (SDBCGR) population is a new high-grade beef cattle breed in China, which is a hybrid of Bohai Black Cattle, Luxi Cattle and Japanese Black Cattle [2]. Its growth performance and meat quality traits are comparable to those of Japanese Kobe beef, and its meat quality meets the international A3 standard, which can fill the shortage of domestic high-grade beef.

Carcass traits controlled by micro-effective polygenes are not only a comprehensive reflection of growth and development of different bodies but also an important economic and breeding index for beef cattle [3,4]. Currently, the main genetic variations are divided into three types: single nucleotide polymorphism (SNP), insertion or deletion (Indel) and copy number variation (CNV). Indel, as a high-throughput genetic marker, is widely utilized in molecular marker assisted selection, animal population genetic analysis, medical diagnosis, etc. [5], and many studies have also provided evidence that Indel polymorphisms are related to characteristic traits among livestock [6]. With the continuous improvement of molecular marker technology, the shortcomings of traditional breeding methods, such as long cycles and unstable inheritance of excellent traits, have been eliminated, and the breeding program for cattle improvement can be better fulfilled. Marker-assisted selection (MAS), is an alternative selection for DNA mutations, which is independent of the microenvironment [7]. Among the three main genetic variations, Indel is the most convenient for detection that does not require special conditions for equipment and technology, and is widely applied in the screening of key economic traits in animals [8]. Therefore, we can continue to improve carcass traits in the SDBCGR population using the MAS method.

The circadian clock is a global regulatory system that produces 24-h cycle rhythm changes in many cellular and body functions [9]. *CRY2*, a circadian regulation factor 2 of cryptochrome, which is a critical component of the circadian core oscillator complex, is necessary for the generation and maintenance of the circadian rhythm [10]. Mutations in clock genes such as *CRY2* can alter circadian rhythms and produce long, short, or irregular phenotypes [11]. As a transcriptional inhibitor, *CRY2* negatively regulates gene transcription and is related to many physiological processes [12]. The circadian rhythm regulates multiple physiological functions, including sleep, body temperature, appetite during exercise and hormone secretion [13]. Interestingly, clock genes can also regulate the physiological functions of skeletal muscle, including muscle growth and maintenance as well as energy metabolism. Numerous rhythm genes in muscle show expression of neurotransmitters at midnight, which corresponds to the peak of physical activity [14]. In addition to shifts in activity, it has been found that disrupted changes in normal circadian gene function may lead to the impairment of metabolic health. The circadian rhythm variations can be associated with the peripheral metabolic effects of the biological clock [15,16]. Moreover, disruptions in circadian rhythm behavior such as shift workers or animal models are more susceptible to metabolic diseases [17]. Meanwhile, the clock mechanism regulates the expression of many metabolic genes in peripheral tissues [18,19]. Circadian clock dysregulation has been associated with many diseases, such as sleep disorders, cancer and metabolic diseases [20,21,22]. Additionally, genetic and epidemiological studies have reported that clock dysbiosis is associated with various adverse metabolic phenotypes [23]. *CRY2*, as an important part of the circadian rhythm, plays a key role in proliferation, DNA repair and DNA damage checkpoint control, regulating important cell cycle progression genes [24,25]. In addition to its effect on metabolism, *CRY2* also affects the growth of the body and alters the generation of muscle cells. To date, there is little research on the *CRY2* gene in relation to carcass traits in cattle.

The final growth performance of animals depends on the interaction of environmental, nutritional and genetic factors [26]. Thus, in terms of genetics, in order to further improve carcass traits in the SDBCGR population, it is necessary to analyze the relationship between gene variations and recorded carcass traits. Hence, the purpose of this study was to detect Indel variations within the *CRY2* gene in the SDBCGR population and further test the correlation between the *CRY2* gene and carcass traits. The results of the current study will provide valuable information for the application of the *CRY2* gene in molecular breeding programs in the SDBCGR population.

## 2. Materials and Methods

### 2.1. Sample Collection and Data Recording

All samples (*n* = 705) from the SDBCGR population come from two similar farms (Shandong, China) and all individuals are similar in body condition, age and feeding conditions (including feed allocation, environment and disease control). In addition, this study used carcass traits such as gross weight, carcass weight (left limb weight and right limb weight), chuck tender, etc. (Figure 1).

Furthermore, after slaughter, farm technicians randomly selected and recorded carcass trait data of males and females, respectively, according to the same standard. Sterile electronic scales were used to weigh the carcass traits such as sirloin and thick flank, and vernier calipers or double calipers were used to measure ribeye morphology and phenotypic characteristics.

### 2.2. Genomic DNA Isolation

Genomic DNA was extracted from the neck muscle tissue samples in accordance with the phenol-chloroform method [27,28]. Firstly, fragmented muscle tissue was lysed with protease K and sodium dodecyl sulfate, and then an equal amount of water-saturated phenol was added to the test tube to be centrifuged to achieve phase separation. The upper aqueous phase was transferred to a new centrifuge tube and then chloroform was added to extract residual phenol from the aqueous phase. DNA was concentrated with anhydrous ethanol. Finally, DNA was washed with 70% ethanol and diluted with water after drying. In addition, the concentration of the DNA samples was tested by Nanodrop 2000 (Thermo Scientific, Waltham, MA, USA). Subsequently, DNA samples were diluted to 20 ng/µL and frozen at −40 °C for further experiments.

### 2.3. PCR Amplification and Genotyping

Eight genetic variation loci were retrieved from the Ensembl database. Then, based on the genetic sequence of the bovine *CRY2* gene, eight pairs of primers (Table 1) were designed using Primer Premier 5 software and synthesized by the Sangon Biotech company (Xi’an, China). Polymorphic fragments were amplified by the polymerase chain reaction (PCR) method using a touch-down program and the PCR amplification procedure was consistent with previous studies [29] (Figure 2). Subsequently, PCR products were detected by 3.0% agarose gels electrophoresis to identify the genotypes of all individuals (*n* = 705). PCR products were then sequenced using Sanger sequencing technology by the Tsingke Biotechnology Company (Xi’an, China).

### 2.4. Statistical Analyses

Genotypic and allelic frequencies were calculated directly based on the different genotypic numbers. Population genetic parameters including heterozygosity (He), homozygosity (Ho), effective allele number (Ne) and polymorphism information content (PIC) were calculated by Nei’s method [30] (Table 2). The correlation between different genotype and carcass traits was analyzed by an independent samples *t*-test with SPSS (Version 25.0, IBM Corporation, New York, NY, USA), and data was presented as the mean ± standard error. *p*-Value < 0.05 was considered to indicate statistical significance. Additionally, Hardy–Weinberg equilibrium (HWE) was measured using the SHEsis online website. The general linear model was as follows: Y*_ijk_* = μ +S*_i_* + G*_j_* + E*_ijk_*, where Y*_ijk_* is the phenotypic value, μ is the overall population mean, S_i_ is the effect of sex, G*_j_* is the genotype, and E*_ijk_* is the random error [31].

## 3. Results

### 3.1. Identification of Indel Variations within the CRY2 Gene

The bovine *CRY2* gene is located on chromosome 15 and consists of 12 exons with a length of 34,781 bp (NCBI https://www.ncbi.nlm.nih.gov/, accessed on 10 December 2020), and based on the eight loci, the bovine *CRY2* gene map was plotted (Figure 3a). The electrophoresis pattern and sequencing map showed that only the *CRY2*-P6 and *CRY2*-P7 loci are polymorphic in the SDBCGR population (Figure 3b–e), and they are deletion and insertion variations, respectively. The *CRY2*-P6 locus had the homozygous insertion (II) genotype and heterozygous (ID) genotype (Figure 3b,c), while the *CRY2*-P7 locus also contained two genotypes: the heterozygous (ID) genotype and (II) homozygous insertion genotype (Figure 3d,e).

### 3.2. Genotypic Frequencies and Population Genetic Parameters Analysis

Based on genetic polymorphisms of the *CRY2*-P6 and *CRY2*-P7 loci, the genotypic frequencies, allelic frequencies and population genetic parameters were calculated. The frequency of the II genotype (0.933) was higher than ID (0.067) within the *CRY2*-P6 locus (Table 3). Similarly, for the *CRY2*-P7 locus, the II genotype had a higher frequency (0.622) (Table 3). In addition, the *CRY2*-P6 locus is consistent with HWE (*p* > 0.05), while *CRY2*-P7 locus deviated from the HWE. PIC value indicated that the *CRY2*-P6 locus had low genetic polymorphism (PIC < 0.25), while the *CRY2*-P7 locus represented medium genetic diversity (0.25 < PIC < 0.5).

### 3.3. Associations between Different Genotypes and Carcass Traits

The correlations between the *CRY2*-P6 locus in the cattle *CRY2* gene and carcass traits of the SDBCGR population were evaluated by an independent samples *t*-test (Table 4). The results of the association analysis reflected that the *CRY2*-P6 locus within the cattle *CRY2* gene was significantly associated with gross weight, back tendon, chuck tender, money tendon, flank steak, triangle flank, ribeye, high rib, beef tenderloin, thick flank, right limbs weight and boneless short ribs in the SDBCGR. Furthermore, individuals with the homozygous genotype displayed a superior phenotype compared to those with heterozygous genotype at the *CRY2*-P6 locus (*p* < 0.05) (Figure 4a,b).

The independent samples *t*-test was also conducted to validate the relationship between the *CRY2*-P7 locus and carcass traits (Table 5). The results showed that there was a significant correlation between the *CRY2*-P7 locus and beef tongue, the length of ribeye, brisket, meat tendon and shoulder clod in male and female SDBCGR. Meanwhile, in males, the II genotype was the dominant genotype (Figure 4c).

### 3.4. Linkage Disequilibrium Analysis of Different Genotypes in the Bovine CRY2 Gene

We performed linkage disequilibrium analysis between the two Indel loci in the intron region of the *CRY2* gene (SHEsis online website, http://analysis.bio-x.cn/, accessed on 10 June 2021). The results exhibited that D’ test value was 0.952 and r^2^ test value was 0.007 (Figure 5). Therefore, according to the D’ test results, there was a strong linkage between the *CRY2*-P6 and *CRY2*-P7 loci; however, r^2^ displayed adverse result. The results of haplotype analysis for *CRY2* generated four haplotypes, and Ip6-Ip7 had the highest frequency (Figure 6).

### 3.5. Analysis of the Combined Effect of the CRY2-P6 and CRY2-P67 Loci on Carcass Traits

Furthermore, the combinatorial genotype analysis between the *CRY2*-P6 and *CRY2*-P7 loci in the SDBCGR population and carcass traits was performed. The genotype with lower frequency (*n* < 3) was excluded from the correlation analysis. In males, the association analysis of three combined genotypes with carcass traits in SDBCGR showed that the three combined genotypes were all significantly associated with several traits (such as money tendon, chuck tender, flank steak, triangle flank and high rib etc.), and combined genotype II-II had a dominant phenotype with significant (*p* < 0.05) differences (Table 6 and Table 7). In females, there were four combined genotypes available for association analysis. Moreover, the combined genotypes were significantly associated with the traits such as chuck tender, thick flank and flank steak, and the combined genotypes II-II and ID-ID had superior (*p* < 0.05) phenotypes. These results further proved that the *CRY2* gene plays an important role in carcass traits of the SDBCGR population (Table 6 and Table 8).

## 4. Discussion

Among the detected loci, the *CRY2*-P6 locus and *CRY2*-P7 locus exhibited polymorphism. Association analysis revealed that the *CRY2*-P6 and *CRY2*-P7 variants in the *CRY2* gene were significantly correlated with SDBCGR population carcass traits (chuck tender, money tendon, flank steak, ribeye, high rib, thick flank, boneless short ribs, beef tongue, etc.). Interestingly, at the *CRY2*-P6 locus, the II genotype was more phenotypically advantageous, and the frequency of the “I” allele was higher compared to the “D” allele. In addition, at the *CRY2*-P7 locus, the II phenotype was more favorable in males. In females, however, it shows a different phenotypic advantage. Previous studies have shown that many physiological processes regulated by circadian rhythm are different between genders [32]. In mammals, females and males have differences in hormones, anatomy and metabolism; in addition, in *CRY2* deficiency, the growth of mice is impaired in a gender-dependent manner, which is due to a change in growth hormone circulation patterns in mutant males [16]. Thus, we speculate that *CRY2* affects body growth in a gender-biased manner. Moreover, we predicted transcription factors at the *CRY2*-P7 locus and found that the FOXA2 transcription factor can specifically bind to the II genotype sequence. FOXA2 contributes to homeostasis in skeletal muscle and is essential for both estrogen and androgen transmission [33]. The PIC value of a marker is equivalent to its ability to detect polymorphisms between individuals in a population and is an indicator of the marker quality in genetic studies; the higher this ability, the greater its value [34]. From the analysis of population parameters, the *CRY2*-P7 locus has a moderate polymorphism (0.25 < PIC < 0.5), which has a high potential genetic value. The *CRY2*-P6 locus is compatible with HWE, while the *CRY2*-P7 locus is not. Possible explanations could be non-random mating, genetic drift or the effects of artificial selection [6].

Our experiment involved many carcass quality traits such as ribeye, thick flank, high rib, beef tongue, chuck tender and boneless short ribs. The clock network is integrated into all major cellular signaling and metabolic pathways, and clock disruptions are expected to affect cell and organism behavior in various ways [23,35,36]. On the one hand, previous studies have shown that melatonin secreted by the pineal gland directly regulated pancreatic insulin secretion in a suprachiasmatic nucleus (SCN) clock-dependent manner [37,38]. It has been proven in studies that the genetic disruption of mouse clock components can induce metabolic diseases, including obesity, by attenuating rhythmic changes in hormone concentrations and expression of metabolic genes [18,19]. Furthermore, *CRY2* deficiency impairs body growth, alters the pattern of circulating growth hormone, and the expression of adipogenic and steroid pathways [16]. Therefore, we hypothesize that deletion of the *CRY2* gene alters hormone secretion and energy metabolism patterns in the organism, thereby affecting body growth. On the other hand, most of the carcass traits involved in this study were related to muscle or bone components. Clock genes can regulate the physiological functions of skeletal muscle, including muscle growth and maintenance as well as energy metabolism. Previous studies have revealed that *CRY2*, rather than *CRY1*, promotes the proliferation of myogenic cells and subsequent formation of myotubes on a circadian basis. Similar differences may occur in chondrocytes [39,40]. Therefore, we suggest that the *CRY2* gene has a function in promoting muscle growth as well as promoting bone metabolism. However, the specific mechanism of how the *CRY2* affects cattle carcass traits remains to be further explored.

Based on the result of the r^2^ value in the LD analysis, there was no linkage between the two loci. Although there was no linkage between the two loci, these variations might have a superposition effect. Considering the large sample size of the SDBCGR population (*n* = 705), the relationship between the combined genotype and carcass traits was analyzed. The results showed that the genotype II-II had better carcass traits. Taking into account the above results, we concluded that the Indel variations of *CRY2* can effectively affect the carcass traits in cattle, which can be used as effective DNA molecular markers for cattle breeding.

## 5. Conclusions

Briefly, we analyzed eight Indel loci within the *CRY2* gene in the SDBCGR population, among which the *CRY2*-P6 and *CRY2*-P7 polymorphisms were polymorphic and significantly associated with carcass traits, and II genotype had the greatest effect on carcass traits at the *CRY2*-P6 locus in SDBCGR population. The results suggest that the *CRY2*-P6 and *CRY2*-P7 variants of the *CRY2* gene can be useful molecular markers for determining carcass traits in cattle. In this field, these molecular markers can be applied in actual production in the future to improve carcass traits in beef cattle and accelerate breeding improvement. However, the possible effect of *CRY2* gene variant on carcass traits needs to be further investigated.

## Figures and Tables

**Figure 1 animals-12-01616-f001:**
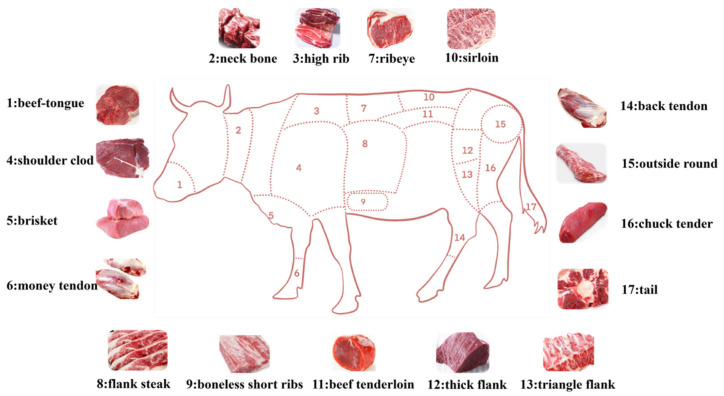
The distribution map of cattle carcass traits involved in this study.

**Figure 2 animals-12-01616-f002:**
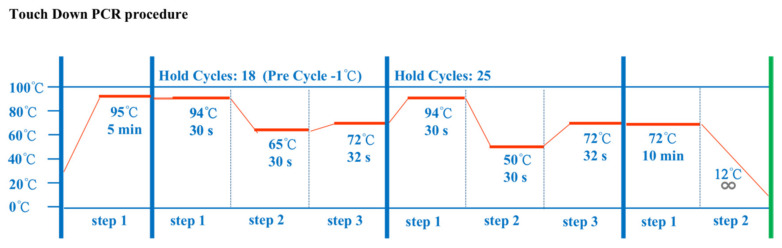
The PCR amplification procedure.

**Figure 3 animals-12-01616-f003:**
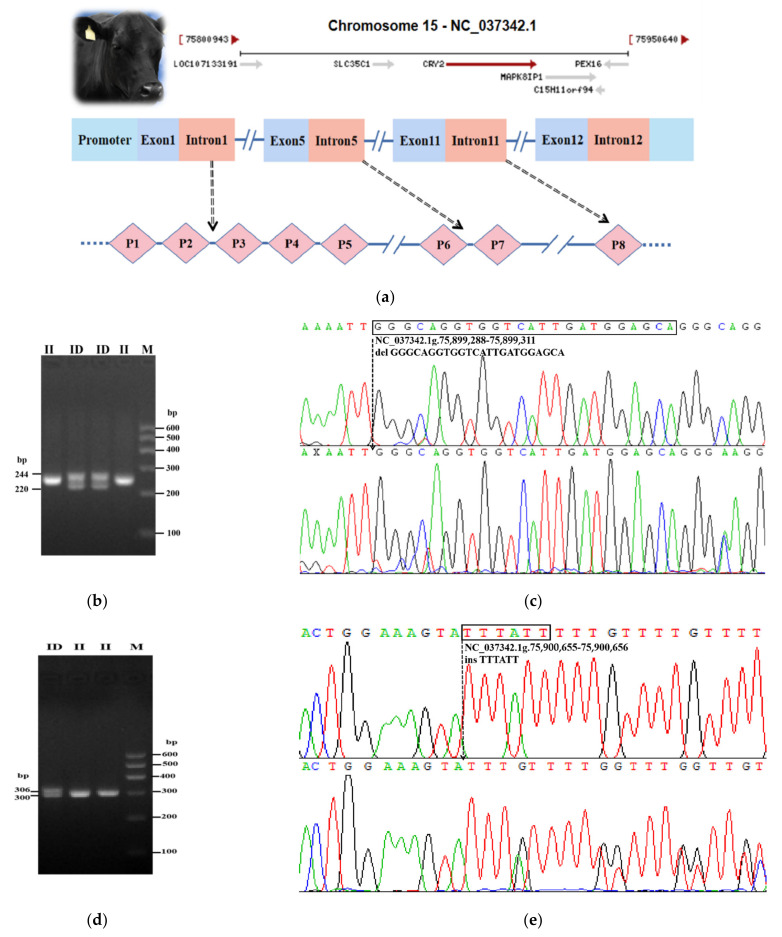
Genotyping of *CRY2*-P6 and *CRY2*-P7. (**a**) The position and gene structure map of *CRY2*; (**b**–**e**) The electrophoresis pattern and sequence chromatograms of the *CRY2*-P6 and *CRY2*-P7 loci in the cattle *CRY2* gene.

**Figure 4 animals-12-01616-f004:**
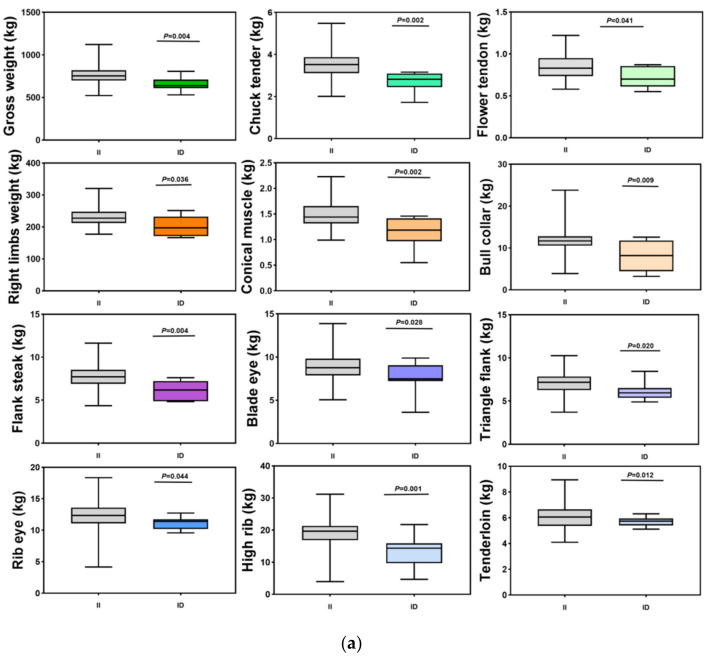
Correlation analysis diagram of carcass traits. (**a**–**c**) Relationship between the *CRY2*-P6 and *CRY2*-P7 loci in the *CRY2* gene and carcass traits in the SDBCGR population. (**a**) *CRY2*-P6, male; (**b**) *CRY2*-P6, female; (**c**) *CRY2*-P7, male.

**Figure 5 animals-12-01616-f005:**
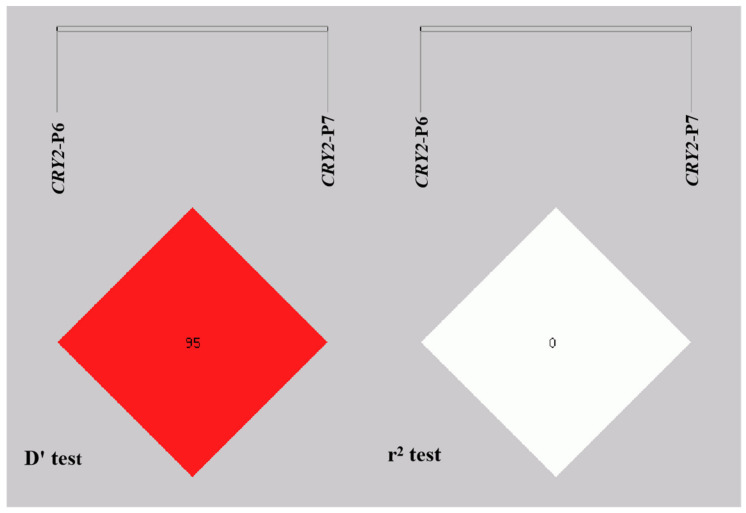
Linkage disequilibrium plot of the two indel loci in the *CRY2* gene.

**Figure 6 animals-12-01616-f006:**
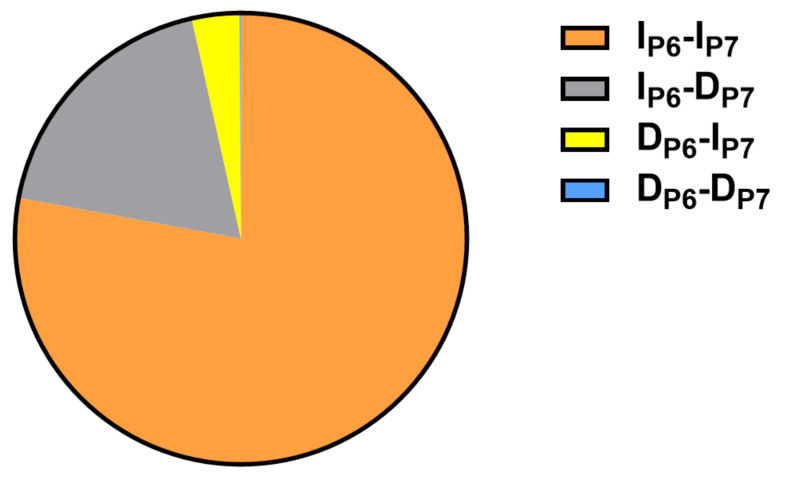
Haplotype frequencies of the *CRY2* gene in the SDBCGR population (I_P6_-I_P7_: 0.779; I_P6_-D_P7_: 0.186; D_P6_-I_P7_: 0.034; D_P6_-D_P7_: 0.001).

**Table 1 animals-12-01616-t001:** PCR primer sequences of the cattle *CRY2* gene.

Names	Primer Sequences (5′ to 3′)	Product Sizes (bp)	Location
P1-5-bp del	F:CCCCACCACCAAAAAAAG R:TGGATGGATGGCATCACC	197/192	intron 1
P2-20-bp ins	F:GCCAGGAGGCCACAATCTG R:ACCATGAGCCATAAACTTCCCA	227/247	intron 1
P3-5-bp del	F:TCCTTCTCCAACGTATGAAAGT R:GAGACACAAGAGATGAGGGTTC	199/194	intron 1
P4-15-bp del	F:CTGGAGTCGTCCCTCATTGG R:AGGTCTCACGATGACCTCCA	245/230	intron 1
P5-8-bp ins	F: TTTCCTCCTGAGAGGACGGT R: CATTCTGGCACGAGCTGAGT	250/258	intron 1
P6-24-bp del	F:AACTGAGAAGGTAAAGCCCTCC R:AGGCAAATACCTCTGCAACC	244/220	intron 5
P7-6-bp ins	F:GGGTTCGAGGAAATAGAGTGAGG R:GCCATCAGGTAGGAAACACCT	306/312	intron 5
P8-13-bp del	F:GACACGGGCCGTCCTAAC R:TCCGGGAAATCAAACGAGGC	227/214	intron 11

**Table 2 animals-12-01616-t002:** Analysis of population genetic parameters.

Population Genetic Parameters	Formula	Purpose
Ho	$H_{o}=\sum_{i=1}^{n}P_{i}^{2}$	Estimation of allele homozygosity
He	$H_{e}=1-\sum_{i=1}^{n}P_{i}^{2}$	Estimation of allele heterozygosity
Ne	$Ne=1/\sum_{i=1}^{n}P_{i}{}^{2}$	Reflecting the interaction between alleles
PIC	$PIC=1-\sum_{i=1}^{m}P_{i}^{2}-\sum_{i=1}^{m-1}\sum_{j=i+1}^{m}2P_{i}^{2}P_{j}^{2}$	Estimation of marker gene polymorphism

Note: The above parameters are calculated using the SHEsis platform (http://analysis.biox.cn, accessed on 8 June 2021).

**Table 3 animals-12-01616-t003:** Genetic diversities of the P6-24-bp deletion and the P7-6-bp insertion of the *CRY2* gene.

Loci	Sizes	Genotypic Frequencies		Allelic Frequencies		HWE	Population Parameters			
		II	ID	I	D	*p*-Values	Ho	He	Ne	PIC
*CRY2*-P6	705	0.933 (*n* = 658)	0.067 (*n* = 47)	0.967	0.033	*p* > 0.05	0.936	0.064	1.069	0.062
*CRY2*-P7	702	0.622 (*n* = 437)	0.378 (*n* = 265)	0.811	0.189	*p* < 0.05	0.694	0.306	1.441	0.259

**Table 4 animals-12-01616-t004:** Relationship between the *CRY2*-P6 locus and carcass traits in SDBCGR.

Carcass Traits	Observed Genotypes (MEAN ± SE)		*p*-Value
	II	ID	
Male (*n* = 195)			
Gross weight (kg)	765.36 ± 7.93 (*n* = 181)	669.73 ± 24.90 (*n* = 11)	0.004
Back tendon (kg)	0.85 ± 0.02 (*n* = 91)	0.72 ± 0.05 (*n* = 6)	0.041
Chuck tender (kg)	3.60 ± 0.07 (*n* = 91)	2.71 ± 0.21 (*n* = 6)	0.002
Right limbs weight (kg)	230.41 ± 3.14 (*n* = 91)	201.00 ± 15.02 (*n* = 5)	0.036
Money tendon (kg)	1.49 ± 0.03 (*n* = 91)	1.15 ± 0.13 (*n* = 6)	0.002
Flank steak (kg)	7.69 ± 0.13 (*n* = 90)	6.13 ± 0.48 (*n* = 6)	0.004
Triangle flank (kg)	7.13 ± 0.13 (*n* = 106)	6.08 ± 0.35 (*n* = 9)	0.020
Ribeye (kg)	12.38 ± 0.19 (*n* = 105)	11.09 ± 0.32 (*n* = 10)	0.044
High rib (kg)	18.72 ± 0.48 (*n* = 106)	13.25 ± 1.56 (*n* = 10)	0.001
Beef tenderloin (kg)	6.10 ± 0.09 (*n* = 103)	5.69 ± 0.12 (*n* = 10)	0.012
Female (*n* = 512)			
Chuck tender (kg)	3.01 ± 0.03 (*n* = 376)	2.79 ± 0.11 (*n* = 33)	0.034
Thick flank (kg)	11.41 ± 0.12 (*n* = 376)	10.38 ± 0.38 (*n* = 33)	0.017
Right limbs weight (kg)	209.08 ± 1.75 (*n* = 377)	196.60 ± 5.38 (*n* = 33)	0.042
Boneless short ribs (kg)	1.24 ± 0.02 (*n* = 372)	1.10 ± 0.06 (*n* = 32)	0.033

Note: (*p* < 0.05) refers to significant differences among the genotypes.

**Table 5 animals-12-01616-t005:** Relationship between the *CRY2*-P7 locus and carcass traits in SDBCGR.

Carcass Traits	Observed Genotypes (MEAN ± SE)		*p*-Value
	II	ID	
Male (*n* = 195)			
Beef tongue (kg)	1.44 ± 0.07 (*n* = 59)	1.28 ± 0.04 (*n* = 33)	0.040
The length of ribeye (cm)	9.54 ± 0.53 (*n* = 8)	7.38 ± 0.46 (*n* = 5)	0.017
Brisket (kg)	9.24 ± 0.34 (*n* = 68)	7.69 ± 0.43 (*n* = 46)	0.005
Female (*n* = 512)			
Meat tendon (kg)	4.74 ± 0.40 (*n* = 186)	6.00 ± 0.47 (*n* = 117)	0.045
Shoulder clod (kg)	1.18 ± 0.03 (*n* = 255)	1.28 ± 0.04 (*n* = 150)	0.021

Note: (*p* < 0.05) refers to significant differences among the genotypes.

**Table 6 animals-12-01616-t006:** Combined genotype frequency of *CRY2* in SDBCGR.

Gender	Combined Genotype Name	Combined Genotype Type	Genotype Frequency
Male	Combined-1	II-II	0.566
	Combined-2	II-ID	0.377
	Combined-3	ID-II	0.047
	Combined-4	ID-ID	0.010
Female	Combined-1	II-II	0.571
	Combined-2	II-ID	0.360
	Combined-3	ID-II	0.055
	Combined-4	ID-ID	0.014

**Table 7 animals-12-01616-t007:** Combined genotype analysis of the *CRY2*-P6 deletion and *CRY2*-P7 insertion loci in SDBCGR (male).

Weight of Carcass Traits	Combined Genotype (MEAN ± SE)			*p*-Value
	II-II	II-ID	ID-II	
Male (*n* = 195)				
Gross weight (kg)	769.97 ^a^ ± 10.18 (*n* = 108)	760.32 ^a^ ± 12.86 (*n* = 72)	671.67 ^b^ ± 30.48 (*n* = 9)	0.033
Money tendon (kg)	1.49 ^a^ ± 0.04 (*n* = 57)	1.48 ^a^ ± 0.39 (*n* = 33)	1.14 ^b^ ± 0.16 (*n* = 5)	0.025
Chuck tender (kg)	3.61 ^a^ ± 0.09 (*n* = 57)	3.52 ^b^ ± 0.10 (*n* = 33)	2.93 ^b^ ± 0.12 (*n* = 5)	0.019
Flank steak (kg)	7.77 ^a^ ± 0.16 (*n* = 57)	7.56 ^a^ ± 0.23 (*n* = 32)	6.39 ^b^ ± 0.49 (*n* = 5)	0.018
Brisket (kg)	9.66 ^a^ ± 0.32 (*n* = 59)	7.69 ^ab^ ± 0.43 (*n* = 45)	7.31 ^b^ ± 1.38 (*n* = 7)	0.001
Triangle flank (kg)	7.42 ^a^ ± 0.16 (*n* = 60)	6.72 ^ab^ ± 0.19 (*n* = 45)	6.10 ^b^ ± 0.40 (*n* = 8)	0.004
High rib (kg)	19.32 ^a^ ± 0.63 (*n* = 61)	17.71 ^a^ ± 0.75 (*n* = 44)	13.79 ^b^ ± 1.71 (*n* = 8)	0.004

Note: Different superscript letters (lower-case letters: *p* < 0.05) refer to significant differences among the genotypes.

**Table 8 animals-12-01616-t008:** Combined genotype analysis of the *CRY2*-P6 deletion and *CRY2*-P7 insertion loci in SDBCGR (female).

Weight of Carcass Traits	Combined Genotype (LSMa ± SE)				*p*-Value
	II-II	II-ID	ID-II	ID-ID	
Female (*n* = 506)					
Gross weight (kg)	688.74 ^a^ ± 4.81 (*n* = 287)	672.02 ^b^ ± 6.13 (*n* = 182)	650.50 ^b^ ± 11.77 (*n* = 28)	679.43 ^ab^ ± 41.60 (*n* = 7)	0.034
Left limbs weight (kg)	210.15 ^a^ ± 2.09 (*n* = 231)	205.75 ^a^ ± 2.92 (*n* = 144)	191.37 ^b^ ± 4.91 (*n* = 26)	211.33 ^ab^ ± 16.61 (*n* = 6)	0.041
Right limbs weight (kg)	210.66 ^ab^ ± 2.16 (*n* = 231)	206.42 ^ab^ ± 2.99 (*n* = 144)	190.73 ^ac^± 5.05 (*n* = 26)	210.92 ^a^ ± 15.80 (*n* = 6)	0.035
Chuck tender (kg)	3.00 ^ab^ ± 0.04 (*n* = 231)	3.02 ^ab^ ± 0.05 (*n* = 143)	2.67 ^ac^ ± 0.11 (*n* = 26)	3.14 ^a^ ± 0.33 (*n* = 6)	0.034
Thick flank (kg)	11.37 ^a^ ± 0.16 (*n* = 231)	11.47 ^a^ ± 0.19 (*n* = 143)	10.02 ^b^ ± 0.38 (*n* = 26)	11.06 ^ab^ ± 0.93 (*n* = 6)	0.037
Shoulder clod (kg)	1.17 ^b^ ± 0.03 (*n* = 230)	1.20 ^b^ ± 0.0 4 (*n* = 144)	1.20 ^b^ ± 0.11 (*n* = 25)	1.99 ^a^± 0.22 (*n* = 6)	3.9 × 105
Flank steak (kg)	6.82 ^a^ ± 0.09 (*n* = 230)	6.75 ^a^ ± 0.11 (*n* = 140)	6.10 ^b^ ± 0.25 (*n* = 26)	7.52 ^a^ ± 0.54 (*n* = 6)	0.029
Outside flat (kg)	18.16 ^ab^ ± 0.49 (*n* = 224)	18.00 ^ab^ ± 0.66 (*n* = 139)	13.75 ^ac^ ± 1.44 (*n* = 26)	17.17 ^a^ ± 1.42 (*n* = 6)	0.041

Note: Different superscript letters (lower-case letters: *p* < 0.05) refer to significant differences among the genotypes.

## Data Availability

Data are available upon request from corresponding author.

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
