# Peer review of "Genetic Variations within the Bovine CRY2 Gene Are Significantly Associated with Carcass Traits"

_animals, 2022, doi:10.3390/ani12131616_

Round 1

Reviewer 1 Report

Li’22 et al-

1.      Line 66- Are there any other genes CRY2, that could play role in circadian regulation.  If so elaborate on them.

2.      Line 155- Can you explain which population parameters are calculated and how they are calculated, any formula or algorhythm. What is purpose of calculating them.

3.      Line 260- Discuss next steps to find mechanism behind differences in gender specific implications of the mutations.

Author Response

Dear Reviewer:

Thank you for your comments, Please check the attachment.

Reviewer 2 Report

Revision of the manuscript entitled “Genetic variations within the bovine CRY2 gene are significantly associated with carcass traits”.

The authors use a PCR analysis to highlight the occurrence of two indel variations in the CRY gene of 705 cattle of the Shandong Black Cattle Genetic Resource (SDBCGR) breed.

The manuscript may be of interest to the field, but some issues need to be addressed. English should be revised by a native english speaker.

Line 33: “deletion mutation”, please delete, please change the word “mutation” to “variation” throughout the tex, your specific variation is an insertion deletion or indel.

Line 50: please rephrase, incorrect logic.

Line 58-59: you cited a paper from 1990, please rephrase.

Line 105: have you genotyped all 705 animals for all the PCR fragments? This must be clearly stated in the materials and methods. Please state how many genotyped animals, and then all the info on animals: from how many different farms? How did you extract DNA? PCR conditions? You should devote a paragraph of the materials and methods section to clearly explain the phenotypic traits collected and how they were measured.

Line 101: “Genomic DNA were extracted from the muscle tissue samples” did you extract genomic DNA from 17 muscle tissue samples from 705 individuals?

Line 104: “frozen at -40 oC for further experiments” which experiments? Please explain.

Line 135-127: please delete, there is nothing with the title.

Line 227-249: Please delete, or insert it in the introduction section.

Line 254: "phenotypically advantageous" what do you mean? Please explain

Line 256: “more favorable” please explain.

Author Response

(The authors gave the same response as above.)

Reviewer 3 Report

I have made a few suggestions for improvement of the manuscript in the attached edited version. Authors should improve on the discussion, conclusions and recommendations for further research.

Author Response

(The authors gave the same response as above.)

Round 2

Reviewer 2 Report

The manuscript has been improved, thank you.